# Method of Food Preparation Influences Blood Glucose Response to a High-Carbohydrate Meal: A Randomised Cross-over Trial

**DOI:** 10.3390/foods9010023

**Published:** 2019-12-25

**Authors:** Caroline Hodges, Fay Archer, Mardiyyah Chowdhury, Bethany L. Evans, Disha J. Ghelani, Maria Mortoglou, Fergus M. Guppy

**Affiliations:** 1School of Pharmacy and Biomolecular Sciences, University of Brighton, Brighton BN2 4GJ, UK; C.Hodges@brighton.ac.uk (C.H.); F.Archer1@uni.brighton.ac.uk (F.A.); M.Chowdhury4@uni.brighton.ac.uk (M.C.); B.Evans6@uni.brighton.ac.uk (B.L.E.); D.Ghelani1@uni.brighton.ac.uk (D.J.G.); M.Mortoglou1@uni.brighton.ac.uk (M.M.); 2Centre for Stress and Age-related Disease, University of Brighton, Brighton BN2 4GJ, UK

**Keywords:** pasta, glycemic index, resistant starch

## Abstract

The aim of this study was to establish the blood glucose response to different cooking methods of pasta. Participants consumed three identical meals in a random order that were freshly cooked (hot), cooled and reheated. Blood glucose concentrations were assessed before, and every 15 min after ingestion of each meal for 120 min. There was a significant interaction between temperature and time (*F*${}_{\left(8.46–372.34\right)}$ = 2.75, *p* = 0.005), with the reheated (90 min) condition returning to baseline faster than both cold (120 min) and hot conditions. Blood glucose area under the curve (AUC) was significantly lower in the reheated (703 ± 56 mmol·L^−1^·min^−1^) than the hot condition (735 ± 77 mmol·L^−1^·min^−1^, *t*${}_{\left(92\right)}$ = −3.36, *p*_bonferroni_ = 0.003), with no significant difference with the cold condition (722 ± 62 mmol·L^−1^·min^−1^). To our knowledge, the current study is the first to show that reheating pasta causes changes in post-prandial glucose response, with a quicker return to fasting levels in both the reheated and cooled conditions than the hot condition. The mechanisms behind the changes in post-prandial blood glucose seen in this study are most likely related to changes in starch structure and how these changes influence glycaemic response.

## 1. Introduction

Dietary carbohydrates are a fundamental constituent of a balanced diet, contributing between 40–70% of energy intake [1], with people relying heavily on staple foods, such as pasta and rice for energy [2,3]. The digestion rates of carbohydrates are determined strongly by the proportions of sugars, starch and fibre present within the carbohydrate. Refined and starchy carbohydrates are readily hydrolysed into their glucose components by pancreatic amylase and brush border enzymes in the small intestine, whereas dietary fibres (including resistant starch, RS) cannot be hydrolysed in the small intestine [4]. Starch that has undergone retrogradation—a process by which the glucose molecules in starch re-associate with each other in an irregular fashion post-gelatinisation [5]—is known to have a high RS content, and therefore will not be digested as effectively in the small intestine and will undergo fermentation by gut bacteria in the large intestine [6]. Therefore, foods with a high RS content produce a lower glycaemic response and contribute towards them having a lower glycaemic index (GI).

Low GI diets have numerous nutritional benefits and may be effective in the management of metabolic syndromes such as obesity and type-2 diabetes mellitus (T2DM), as they produce a lower post-prandial blood glucose response [7], as well as effects on cardiometabolic and inflammatory markers [8]. There is a body of evidence supporting the link between RS digestion and post-prandial hyperglycaemia and insulinaemia, because RS-rich foods are harder to digest. The preparation of food can also influence the GI properties of a meal, with retrogradation of starch by cooling and reheating, increasing the amount of RS, particularly type 3, present [9]. However, there is a lack of published evidence regarding humans that supports the effect of cooking on glycaemic response to a high carbohydrate meal.

The aim of this study was to establish the blood glucose response to a pasta meal that was cooked and eaten hot; cooked and eaten after cooling or cooked and cooled and then reheated. We hypothesised that the cooked and eaten hot meal would produce the highest glycaemic response when compared with the other methods of preparation because of its disorganised, amorphous structure and swollen starch granules produced by gelatinization, with the cooked, cooled then reheated meal causing the lowest glycaemic response due to its repeated retrogradation from being cooked, cooled and then reheated.

## 2. Materials and Methods

### 2.1. Participants

Forty-five volunteers (age = 20–24 years) took part in this study and were informed of the experimental protocol both verbally and in writing before giving informed consent. The study protocol was approved by the School of Pharmacy and Biomolecular Sciences Research Ethics Panel (approval number: PABS-REP-2017-05).

### 2.2. Experimental Conditions

Participants undertook each experimental condition in an unblinded random order decided by a random number generator, with each experimental visit separated by a minimum of 48 h. Participants were instructed to refrain from performing any strenuous physical activity for 2 days prior to each experimental visit and attended the laboratory after an overnight fast. Participants were excluded from the study if fasting plasma glucose exceeded 7 mmol/L during their first visit to the laboratory. No participants had fasting plasma glucose concentrations in excess of 7 mmol/L in their first experimental visit, although 15 participants had values in excess of 5.6 mmol/L on at least one visit, while 5 participants had fasting plasma glucose values in excess of 5.6 mmol/L on two visits.

### 2.3. Pasta Preparation

Three different preparations of white fusilli pasta (Asda Stores Ltd., Leeds, UK) with a simple Tomato and Basil Stir-in pasta sauce (Dolmio^®^, Mars Inc., Slough, UK) were tested in this study: hot, cold and reheated. Each participant was given 100 g (dry weight) of pasta, which was cooked in water for 20 min [10], at a ratio of 566 mL of water to 100 g of pasta, with 100 g of pasta sauce. The hot pasta meal was freshly cooked, the cold pasta meal was cooked and chilled for 24 h overnight in a refrigerator at 4 °C in a sealed plastic container, while the reheated pasta meal followed the same treatment as the cold condition but was then reheated on the day of the experiment for 3 min in a 750 W microwave (Proline SM18) on the high setting, with stirring every minute. Each subject was provided with 250 mL of water with their meal, which they were asked to ingest within 15 min [11].

### 2.4. Blood Glucose Responses

Capillary blood samples were collected by the participant by single use lancet from the fingertip before, and every 15 min for 120 min after ingestion of the meal. Whole blood glucose concentrations were measured using an automatic analyser (Accu-Chek Performa Blood Glucose Meter, Roche Diagnostics, Basel, Switzerland).

### 2.5. Calculations and Data Analysis

Area under the glucose curve (AUC) was calculated using the conventional trapezoid rule. Blood glucose response was analysed using a 2-way repeated measures ANOVA, and area under the curve was analysed using a 1-way repeated measures ANOVA (Jamovi v 0.9.5.12 [12]). Assessment of sphericity using Machualy’s test indicated that there was a violation of this assumption for the both the time and treatment x time interaction in blood glucose response, therefore Greenhouse–Geisser corrections were applied. Pairwise comparisons were conducted using a Bonferonni post-hoc correction (*p*_bonferonni_). Data shown is mean ± standard deviation unless otherwise stated, with significance accepted if *p* < 0.05.

## 3. Results

### 3.1. Preparation Method

There was a significant effect of the preparation method of the pasta (*F*${}_{\left(2–88\right)}$= 4.40, *p* = 0.015), with mean 2 h blood glucose concentration significantly lower in the reheated condition (5.78 ± 0.91 mmol/L) than in the hot condition (6.03 ± 1.02 mmol/L, *t*${}_{\left(88\right)}$ = 2.94, *p*_bonferroni_ = 0.013). There were no differences in mean 2 h blood glucose concentrations between cold pasta (5.94 ± 0.95 mmol/L) and either hot (*t*${}_{\left(88\right)}$ = 1.10, *p*_bonferroni_ = 0.820), or reheated (*t*${}_{\left(88\right)}$ = 1.83, *p*_bonferroni_ = 0.210) pasta.

### 3.2. Time

Pasta ingestion caused significant increases in blood glucose regardless of the preparation method (*F*${}_{\left(3.06–134.81\right)}$ = 59.97, *p*< 0.001), with significantly increased blood glucose at each time point than the previous time point up to 30 min (Table 1).

### 3.3. Preparation Method and Time Interaction

There was a significant interaction between preparation method and time (*F*${}_{\left(8.46–372.34\right)}$ = 2.75 *p* = 0.005). The reheated condition saw a faster return to baseline than both cold and hot conditions, with the reheated condition seeing a return to baseline values within 90 min, compared with 120 min in the cold condition, and the hot condition not returning to baseline by the end of the 2 h period (Table 1). However, there were no differences at any time point between any condition (Figure A1).

### 3.4. Area Under the Curve

There was a significant effect of preparation method on AUC (*F*${}_{\left(2–92\right)}$ = 6.19, *p* = 0.003). Reheated pasta had a significantly lower area under the curve (703 ± 56 mmol·L^−1^·min^−1^) than the hot condition (735 ± 77 mmol·L^−1^·min^−1^, *t*${}_{\left(92\right)}$ = −3.36, *p*_bonferroni_ = 0.003), with no significant difference with the cold condition observed (722 ± 62 mmol·L^−1^·min^−1^, *t*${}_{\left(92\right)}$ = −2.07 *p*_bonferroni_ = 0.123). There was also no significant difference between the cold and hot condition (*t*${}_{\left(92\right)}$ = −1.29, *p*_bonferroni_ = 0.601, Figure 1).

## 4. Discussion

This study aimed to examine the effect of cooking methodology of pasta on post-prandial blood glucose, and found that both cooled, and reheated pasta, were associated with a faster return to baseline blood glucose, than the hot condition, while reheated pasta also showed significantly reduced blood glucose AUC, compared with freshly cooked pasta.

To our knowledge, the current study is the first to show that reheating pasta causes changes in post-prandial glucose response, with a quicker return to fasting levels in both the reheated and cooled conditions, than the hot condition. The mechanisms behind the changes in post-prandial blood glucose seen in this study are most likely related to modifications of starch structure and the subsequent influence on the glycaemic response. Studies in potatoes, noodles, rice and lentils indicate that cooking and cooling changes the amount of RS present [6,13,14,15,16,17,18] changing the digestibility of these foods [19,20]. This alteration in chemical structure, in conjunction with changes in amylopectin and amylose crystallisation, may contribute to the indigestibility of starch [21,22]. These retrograded RS molecules form tight structures stabilised by hydrogen bonds [5]. This modified structure means that digestive enzymes (e.g., α-amylase) less effectively digest starch [23] resulting in food with a lower GI [24].

A significantly faster return to baseline in the reheated condition, than in the hot condition was also shown, which contributed to the decrease in AUC after ingestion of the reheated pasta. Sonia et al. [25] found lower blood glucose levels and AUC after consumption of reheated rice, compared with control rice, and suggested this was most likely attributable to higher RS, which would decrease the available carbohydrate content. A similar result was obtained by Lu et al. [26], when comparing freshly cooked white rice with reheated cold-stored parboiled rice and in vitro digestion studies showed that freshly cooked warm rice was digested more rapidly than cold-stored and reheated rice and minced-reheated parboiled rice was more resistant to digestion [27].

The authors suggest that the amorphous ordered crystalline state produced after cooling, persists on reheating. In this study, only one heating–cooling cycle was applied, however, a review by Boers et al. [28], indicated that multiple heating–cooling cycles of rice can achieve a higher RS content. Retrograded amylopectin is generally thought to melt upon reheating, due to the low melting point (46–65 °C) of these crystallites [29], but during reheating, the amylopectin crystallites may retain some of the associating forces, which could partially explain why pressure parboiling of rice elicits a relatively low GR. Some amylose–lipid complexes have a melting temperature above 100 °C and are not melted during the cooking process, resulting in higher levels of RS [30].

Studies on potatoes [31], and maize porridge [32], similarly observed a lower GR and RS respectively upon consumption after reheating, than when freshly prepared. Darman–Djoulde et al. [32] suggested the storage-reheating process increased RS by promoting the interaction of starch with other components such as proteins, lipids or itself. Conditions known to decrease the digestibility of starch and subsequent GR responses, include those which increase lipid–amylose formation [21]. Robertson [9] states that while common food preparation methods, including reheating, are thought to be involved in increasing RS, the mechanism by which this is achieved is unclear.

Most research agrees that cooling carbohydrates results in retrogradation and subsequent RS formation. However, there is considerably less agreement when looking at the effect of reheating which, above 65 °C, is thought to reverse retrogradation by breaking some of the amylopectin crystallites, thereby decreasing RS and rendering the food more susceptible to digestion. However other studies, including ours, did not observe this effect, and suggest upon reheating, the retrograded amylopectin crystallites may not be converted into a digestible form. Temperatures above 145 °C are required to remove crystallinity of retrograded amylose, a temperature which exceeds the range used for processing of starchy foods. This implies that retrograded amylose, once formed, may retain its crystallinity following re-heating [33]. What has yet to be determined is why reheated starches may contain more RS, than cooled.

The decreased AUC resulting from reheating is significant because reducing post-prandial glucose fluctuations has several benefits, such as the reduction of inflammation and oxidative stress, [34]. Schisano et al. [35], reported that exposure of cultured endothelial cells to oscillating glucose concentrations was more deleterious than constant high glucose exposure and induced a metabolic memory after glucose normalisation, as well as causing greater apoptosis. In non-diabetic lean patients, reduced post-prandial NFκB activation in white blood cells resulted from meals which elicited a flatter glycaemic response [36]. In patients with T2DM glycaemic variability is implicated in coronary artery disease [37]. For example oxidative stress is activated by glycaemic fluctuations [38] and incremental glucose peaks have been shown to correlate with carotid intima-media thickness, which is a surrogate marker for atherosclerosis [39].

In conclusion, although it is evident that the cooking methodology of pasta influences post-prandial glucose response, with a faster return to baseline in both cooled and reheated pasta, as well as reduced AUC following reheating, further work is needed to understand the mechanisms driving these changes and to ascertain if alterations in chemical structure is the primary factor influencing post-prandial glucose response.

## Figures and Tables

**Figure 1 foods-09-00023-f001:**
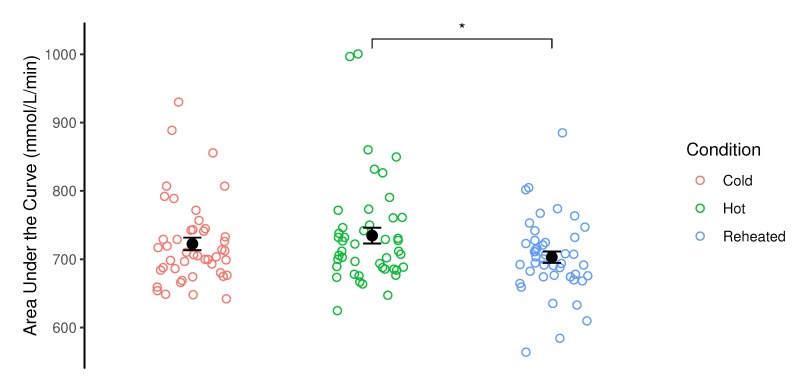
Area under the curve for post-prandial glucose response to three different carbohydrate meal preparations. Data shown is mean ± standard error alongside individual responses. * denotes significant difference between groups (p<0.05).

**Table 1 foods-09-00023-t001:** Post-prandial glucose response.

Time (mins)	0	15	30	45	60	75	90	105	120
Cold	5.16	5.68 ${}^{A}$	6.63 ${}^{B,C}$	6.58 ${}^{B}$	6.09 ${}^{B}$	6.04 ${}^{B}$	5.72 ${}^{A}$	5.72 ${}^{A}$	5.63
	[0.57]	[0.94]	[1.08]	[1.17]	[0.91]	[0.91]	[0.66]	[0.58]	[0.65]
Hot	5.07	5.91 ${}^{B}$	7.10 ${}^{B,C}$	6.68 ${}^{B}$	6.26 ${}^{B}$	5.90 ${}^{B}$	5.97 ${}^{B}$	5.82 ${}^{B}$	5.58 ${}^{B}$
	[0.70]	[0.76]	[0.95]	[1.00]	[1.23]	[0.77]	[0.78]	[0.78]	[0.67]
Reheated	5.05	5.82 ${}^{B}$	6.94 ${}^{B,C}$	6.47 ${}^{B}$	5.85 ${}^{B}$	5.57 ${}^{A}$	5.56	5.46	5.33
	[0.44]	[0.71]	[0.92]	[0.97]	[0.93]	[0.74]	[0.62]	[0.63]	[0.59]

Data shown is mean [standard deviation]. All *n* = 45. A/B significantly different from baseline (*p* < 0.05/*p* < 0.001). C significantly different from previous time-point (*p* < 0.05).

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
