# Peer review of "Method of Food Preparation Influences Blood Glucose Response to a High-Carbohydrate Meal: A Randomised Cross-over Trial"

_foods, 2019, doi:10.3390/foods9010023_

Round 1

Reviewer 1 Report

The manuscript presented by Hodges et al. compared the postprandial response using different pasta cooking process.

The manuscript is well designed and explained. However, I´m afraid that the originality of this work is quite low. For example, back in 1986, Wolove et al. already discussed the degree of cooking on the glucose response (https://doi.org/10.2337/diacare.9.4.401). In this sense, I recommend to the authors to include and discus this reference.

In general, this paper should be submitted as a short communication.

Author Response

Thank you for praising the design and explanation of the study.  We felt that the study was different from the Wolover et al (1986) study, which examined different cooking times of pasta rather than what happens when pasta is cooked, cooled and reheated, to not include this reference within the present manuscript.  We have however, expanded the discussion including more work looking at the cooking effects on glycemic response in this population.

Reviewer 2 Report

Hodges and coworkers have undertaken a randomised, crossover trial in healthy participants in order to determine if the method of preparation of carbohydrate-containing foods can influence post-prandial blood glucose responses.

Major points:

I completely understand why you did it, but it is always very disappointing to see yet another study done in young, healthy volunteers. Please consider doing a similar study in 1) children with T1DM and 2) people with T2DM. I know this results in wider confidence intervals and thus requires a larger sample size, but a lot of recent meta-analyses have shown very different effects in people with diabetes, impaired glucose tolerance and normal blood glucose concentrations, meaning that your study may not be of that much relevance to people for whom glycaemic index is most important. There is a conflation in the manuscript between resistant starch and low GI diets. Low GI diets include mechanisms other than resistant starch to achieve the lowering of glycaemic index. Also, no mention is made of the various kinds of resistant starch and how they may differ in their effects. You refer to resistant starch in general, but really you are talking about RS3. Whether you can extrapolate to RS1, 2 and 4 remains to be seen. It was surprising to see you cooked the pasta for 20 minutes – this is far longer than the normal preparation time for pasta. Fusilli pasta normally takes between 9 and 11 minutes to cook. Please comment on any effect this might have had. Did you publish a protocol? If so, please link to this. You should include a table with baseline data on the participants You should include a trial flow diagram You should include the registration number of the clinical trial It is not clear at all what you are measuring in your results, especially under “Preparation Method”. You should state what outcome you are reporting on. Is it peak? Two hour? I cannot work out what this result means. Please discuss your hypothesis explaining the observed difference between cold and reheated pasta. In theory the retrogradation should be complete after 24 hours at 4C. So if anything you might expect this to decrease upon reheating, or at least that it would remain the same. Do you think this effect is real? Or a chance observation?

Minor points:

You should identify your study as an RCT in the title and remove the full stop from the end of the title. THANK YOU for using “compared with” rather than “compared to”!! Low GI diets have benefits for people with obesity and diabetes beyond their effects on the post-prandial response. Please mention this and include appropriate references. Also references 7 and 8 refer to resistant starch diets, not low GI diets, per se. You should find a specific meta-analysis to use instead. You should state in the title if this is single blind or double blind – I realise it’s impossible to blind the cold vs hot pasta, but not fresh vs reheated pasta. You have stated the method of randomisation but did not state how you maintained blinding (if at all) among the participants and personnel, both to the trial itself and to the allocation. Who did the randomisation? Who did the allocation? How was allocation concealed? You should state how you arrived at your sample size – what was the expected difference and how did you calculate this? In your AUC calculations, did you include or ignore any negative area (i.e. blood glucose concentrations lower than baseline?

Author Response

Hodges and coworkers have undertaken a randomised, crossover trial in healthy participants in order to determine if the method of preparation of carbohydrate-containing foods can influence post-prandial blood glucose responses.

Major points:

I completely understand why you did it, but it is always very disappointing to see yet another study done in young, healthy volunteers. Please consider doing a similar study in 1) children with T1DM and 2) people with T2DM. I know this results in wider confidence intervals and thus requires a larger sample size, but a lot of recent meta-analyses have shown very different effects in people with diabetes, impaired glucose tolerance and normal blood glucose concentrations, meaning that your study may not be of that much relevance to people for whom glycaemic index is most important.

The authors acknowledge the comment from the reviewer and agree that this study would be valuable if conducted in the populations discussed.  We hope that this is the first study in a series of work, and also hope that we will be able to access these populations in the future as this would provide a logical next step.  However, we also feel that understanding this relationship in healthy individuals is also important as a potential mechanism for allowing people to manage diet in order to potentially help improve lifestyle choices in this population.

There is a conflation in the manuscript between resistant starch and low GI diets. Low GI diets include mechanisms other than resistant starch to achieve the lowering of glycaemic index. Also, no mention is made of the various kinds of resistant starch and how they may differ in their effects. You refer to resistant starch in general, but really you are talking about RS3. Whether you can extrapolate to RS1, 2 and 4 remains to be seen.

We appreciate the reviewer’s insight into the conflation between RS and GI, we have attempted to clarify this where appropriate. For brevity, we have avoided too much discussion about specific sub-types of RS as we felt this was beyond the scope of this paper.

It was surprising to see you cooked the pasta for 20 minutes – this is far longer than the normal preparation time for pasta. Fusilli pasta normally takes between 9 and 11 minutes to cook. Please comment on any effect this might have had.

The pasta was cooked in a microwave, so required slightly longer cooking time.    We have added a reference to the methods section that showed that similar cooking was achieved in 20 minutes microwave cooking to traditional cooking (8 minutes) (Cocci et al, 2008 [10]). 

Did you publish a protocol? If so, please link to this. You should include a table with baseline data on the participants You should include a trial flow diagram. You should include the registration number of the clinical trial.

We did not publish a protocol prior to undertaking the study, and this trial was not a registered clinical trial so we do not have a registration number.

It is not clear at all what you are measuring in your results, especially under “Preparation Method”. You should state what outcome you are reporting on. Is it peak? Two hour? I cannot work out what this result means.

Thank you for pointing out this confusion, the preparation method section of the results has been updated to include clarification as to the data presented here which is mean 2-hour glucose response.

Please discuss your hypothesis explaining the observed difference between cold and reheated pasta. In theory the retrogradation should be complete after 24 hours at 4C. So if anything you might expect this to decrease upon reheating, or at least that it would remain the same. Do you think this effect is real? Or a chance observation?

We accept there are many studies which indicate retrogradation is complete after cooling for 24 hours, but do not show a decreased GR upon reheating.

However, a number of other studies have shown an effect similar to ours when reheating rice, potatoes and porridge, we have now expanded the discussion to include this.

Minor points:

You should identify your study as an RCT in the title and remove the full stop from the end of the title.

The title has been amended as suggested.

THANK YOU for using “compared with” rather than “compared to”!!

Thank you for the positive review comment.

Low GI diets have benefits for people with obesity and diabetes beyond their effects on the post-prandial response. Please mention this and include appropriate references. Also references 7 and 8 refer to resistant starch diets, not low GI diets, per se. You should find a specific meta-analysis to use instead.

We have added in 2 meta-analyses that refer to some of the benefits of a low GI diet [Ojo et. al. 2018/2019]

You should state in the title if this is single blind or double blind – I realise it’s impossible to blind the cold vs hot pasta, but not fresh vs reheated pasta. You have stated the method of randomisation but did not state how you maintained blinding (if at all) among the participants and personnel, both to the trial itself and to the allocation.

The allocation was unblinded, with participants aware of which condition they were undertaking. This has been clarified in the methods.

Who did the randomisation? Who did the allocation? How was allocation concealed? You should state how you arrived at your sample size – what was the expected difference and how did you calculate this?

This work was conducted as part of 5 undergraduate projects, with each student recruiting 9 participants giving a total sample size of 45. No a priori power calculations were conducted to determine sample size.  Previous unpublished work indicated that effect sizes of d= 0.25 were observed with similar protocols in n=10 individuals, although this was not used to calculate sample size.

Randomisation and treatment allocation for each individual sub-set of 9 participants was done by FA, MC, BLW, DJG and MM under the supervision of CH.

In your AUC calculations, did you include or ignore any negative area (i.e. blood glucose concentrations lower than baseline?

Area under the curve was calculated to include any negative areas where blood glucose concentration dropped below baseline.

Reviewer 3 Report

-line 7, Throughout the paper, the consistent grammar errors were found. When comparative adjectives are used to compare differences between two objectives, it should be as following: comparative adjective + "than", not compared with.

-lines 30-32, the sentence should be re-written because “the link between RS digestion and postprandial hyperglycemia and insulinemia” is not well related to “have a lower GI”.

-For results, participants’ demographic information should be included.

-Table 1 and Fig 1 provided the same info. Either of them should be removed.

-Fig 1: remove individual responses and include SD/SE at each time point.

-Fig 2: make a table for AUC results, not a fig.

-data analysis should be re-done and organized.

Write discussion more thoroughly.

Round 2

Reviewer 1 Report

Dear authors,

Thanks for the revision. I still think that based on the experimental design, impact, significance of the results and conclusions, the manuscript should be submitted as a short communication. 

Kind regards.

Author Response

We appreciate the reviewers feedback on the corrections made.  We will leave the decision as to whether this article is better suited to a short communication to the Editorial Team.

Reviewer 3 Report

-Figure 1, add significant differences to Figure 1.

-Display 3 meal preparations in the same order throughout the manuscript.

-Fig A1, several participants’ fasting blood glucose levels for 3 treatments at baseline are higher than 5.6 mmol/L, which is a normal fasting blood glucose level cutoff. Based on the fasting blood glucose levels, some participants cannot be considered “healthy”.

Also, for this cross-over study design, it is not sure if the same participants had higher fasting blood glucose level than normal for all three treatments or if randomly some participants showed higher fasting blood glucose level than normal for 1-2 treatments.

-Demographic info and baseline of participants are significant to include in the manuscript for any human studies. Also, the info would be valuable to reviewers to check if participants truly meet the eligibility criteria.

-No eligibility criteria described in the methods.

-lines 118 and 199,  “significantly lower blood glucose concentration in the reheated condition, than in the hot

119 condition was also shown,” is not true because there were no significant differences in blood glucose levels at any time points in Table 1.  Same for the sentence in lines 151-152, “flattening the glucose response

152 by reducing peak rise, reduces post-prandial glucose fluctuations”.
